# Aerosol light absorption from optical measurements of PTFE-membrane filter samples: sensitivity analysis of optical depth measures

Apoorva Pandey[1], Nishit J. Shetty[1], and Rajan K. Chakrabarty[1,2]

[1]Center for Aerosol Science and Engineering, Department of Energy, Environmental and Chemical Engineering, Washington University in St. Louis, St. Louis, MO 63130, USA

[2]McDonnell Center for the Space Sciences, Washington University in St. Louis, St. Louis, MO 63130, USA

*Correspondence to*: Rajan K. Chakrabarty (chakrabarty@wustl.edu)

**Abstract.** Mass absorption cross-section (MAC) measurements of atmospherically-relevant aerosols are required to quantify their effect on Earth's radiative budget. Estimating aerosol light absorption from transmittance and/or reflectance measurements through filter deposits is an attractive option because of ease of deployment in field settings, low cost, and the ability to revisit previously analysed samples. These measurements suffer from artifacts that depend on a given filter measurement system and aerosol optics. Empirical correction algorithms are available for commercial instruments equipped with optically-thick fiber filters, but optically-thin filter media have not been characterized in detail. Here, we present empirical relationships between particle light absorption optical depth–measured using multi-wavelength integrated photoacoustic spectrometers–and filter optical depth measurements for polytetrafluoroethylene (PTFE) membrane filter samples of carbonaceous aerosols generated from combustion of diverse biomass fuels and kerosene (surrogate for fossil-fuel combustion). Through radiative transfer modeling, we assessed the suitability of three measures of filter-based optical depth for robustly describing particulate-phase light absorption over a range of single scattering albedo (SSA) values: (1) $OD_s$–a measure of transmission of the fraction of incident radiation that is not backscattered by the filter system–utilizes transmittance and reflectance of the sample side of the filter, (2) the commonly-used $OD_c$ uses transmittance and reflectance of the clean side of the filter, and (3) *ATN* or the Beer-Lambert attenuation. Modeling results were also experimentally validated, with $OD_s$ showing the least variability around the mean in this multi-dimensional parameter space. We establish a simple, wavelength-independent formulation for calculating aerosol MAC and absorption coefficients from measurements of $OD_s$. We find the ratio between in-situ particulate absorption optical depth and $OD_s$ to be inversely proportional to aerosol SSA. Our findings underscore that $OD_s$ is a better optical depth measure than $OD_c$ for applying appropriate correction factors when estimating particle phase light absorption from filter-based techniques.

## 1 Introduction

Aerosol light absorption affects the radiative balance of the Earth's atmosphere through direct and indirect mechanisms (Bond et al., 2013; Kanakidou et al., 2005; Ramanathan et al., 2001). The light absorption metric relevant to climate modelers–mass absorption cross-section (MAC)–depends on the size, shape and composition of the aerosols (Andreae and

Gelencsér, 2006; Bond and Bergstrom, 2006; Moosmüller et al., 2009). This property has a complex relationship with the emission source, especially for carbonaceous aerosols (Andreae and Gelencsér, 2006; Bond and Bergstrom, 2006; Chakrabarty et al., 2010). The canonical MAC value for "pure" black carbon aerosols, associated with fossil fuel emissions, is 7.5 $m^2g^{-1}$ at 550 nm (Bond et al., 2013). MAC values for absorbing organic carbon, typically released from biomass

burning, strongly vary with combustion phase, and can range ~0.01-1 $m^2g^{-1}$ at 550 nm (Laskin et al., 2015).

A first-principle method of measuring contact-free aerosol light absorption is photoacoustic spectroscopy, which employs lasers at selected wavelengths to heat the aerosols, thereby producing a detectable pressure signal (Arnott et al., 1999). Absorption can also be estimated as the difference between *in-situ* measurements of extinction and scattering (Schnaiter et al., 2005; Sheridan et al., 2005). Alternatively, a commonly adopted technique for estimating light absorption uses

measurements of transmittance and/or reflectance for aerosol particles collected on a filter substrate. Instruments based on this technique, including  the aethalometer (Hansen et al., 1984) and the Particle Soot Absorption Photometer or PSAP (Virkkula et al., 2005), facilitate  semi-continuous sampling of particles and produce time-averaged bulk absorption measurements. Particles may also be collected on a filter substrate and analyzed for their absorption using standalone spectrophotometers (Martins et al., 2009; Pandey et al., 2016; White et al., 2016; Zhong and Jang, 2011).

Filter-based measurements are attractive because of their ease of deployment in field settings and low cost, but they suffer from several artifacts. Particles embedded in a multiple-scattering medium experience a larger optical path length than in their native suspended state, leading to the appearance of enhanced light absorption (Bond et al., 1999; Clarke, 1982; Gorbunov et al., 2002). This is referred to the as the multiple scattering artifact and depends on the choice of filter medium. A higher loading of absorbing aerosols can diminish the effect of multiple scattering, inducing an aerosol dependent loading

artifact (Arnott et al., 2005; Weingartner et al., 2003). Highly scattering aerosols could enhance multiple scattering and lead to increased backscatter, which leads to an overestimation of absorption (Lack et al., 2008; Weingartner et al., 2003). These artifacts have been evaluated for several commonly used filter-based instruments, such as those aforementioned, by comparing their measurements with contact-free aerosol light absorption measurements or using reference materials with known optical properties. Typically, correction algorithms for these artifacts are formulated as functions of some

combination of filter and aerosol properties (Arnott et al., 2005; Collaud Coen et al., 2010; Virkkula, 2010; Weingartner et al., 2003) and are specific to a given measurement system.

In many field settings, aerosol samples are collected on polytetrafluoroethylene (PTFE) membrane filters (commonly known as Teflon filters) for inferring ambient or near-source particulate mass concentrations using gravimetric analysis (Koistinen et al., 1999). Major aerosol monitoring networks, such as the Interagency Monitoring of PROtected Visual Environments

(IMPROVE) network (Chow et al., 2010; Solomon et al., 2014), the Chemical Speciation Network (CSN) (Solomon et al., 2014) and the Surface PARTiculate mAtter Network (SPARTAN) (Snider et al., 2015), collect particle samples on Teflon filters for gravimetric and elemental measurements. PTFE filters are chemically inert and unlike quartz fiber filters, present a very low surface area for organic vapor adsorption (Kirchstetter et al., 2001; Vecchi et al., 2014). Correction schemes developed for instruments that use fiber filters (like the PSAP and aethalometer) cannot be applied to transmittance and/or

reflectance measurements on PTFE filters. A previous study on the artifacts associated with this filter type used a reference material and provided a constant multiple scattering correction factor for optical loadings smaller than a certain threshold (Zhong and Jang, 2011). Another recent study (White et al., 2016) proposed a theory-based model to calibrate attenuation measurements for Teflon filter samples and applied this new model to a historical dataset from IMPROVE network. They found that the reevaluated absorption values for the PTFE samples were well-correlated with thermo-optical elemental carbon (EC) measurements for co-located quartz fiber filters.

In this work, we generated carbonaceous aerosols with varying physicochemical properties from the combustion of biomass fuels and kerosene. Kerosene combustion was used as a surrogate for fossil fuel burning, which is linked with soot or EC emissions (Andreae and Gelencsér, 2006; Bond et al., 2013). The combustion of wildland- and fuel-biomass is implicated in emissions of EC as well as light absorbing organic carbon (LAOC) (Andreae and Gelencsér, 2006; Chakrabarty et al., 2010; Chen and Bond, 2010). EC is known to absorb light throughout the visible and ultraviolet (UV) wavelengths, while LAOC absorbs preferentially in the near-UV and UV regions (Andreae and Gelencsér, 2006; Bond and Bergstrom, 2006; Kirchstetter et al., 2004; Sun et al., 2007). Therefore, we measured *in-situ* and contact-free aerosol light absorption and scattering coefficients using integrated photoacoustic-nephelometer (IPN) spectrometers operated at three wavelengths - 375, 405 and 532 nm. Co-located with these measurements was a sampling system to collect particles onto Teflon membrane filters. Subsequent filter optical measurements, using UV-visible (UV-vis) spectrophotometer, were performed. Observed empirical relationships between particle light absorption and filter optical depth measures were established in conjunction with predictions from a one-dimensional (1-D) two-stream radiation transfer model.

## 2 Methods

### 2.1 Two-stream radiative transfer model

A 1-D two-stream radiative transfer framework for multiple scattering in absorbing media was developed in Bohren (1987)–widely known as the Kubelka-Munk theory (Kubelka, 1948)–and subsequently discussed in relation to aerosol-filter systems in several studies (Arnott et al., 2005; Clarke, 1982; Gorbunov et al., 2002; Petzold and Schönlinner, 2004). Solving a radiation balance for an aerosol-laden filter medium yields the following expressions for transmittance ($T_l$) and reflectance ($R_l$), respectively:

$$T_l = \frac{2}{[2K - \omega_l(1-g_l)\sinh(K\tau_{e,l}) + 2K\cosh(K\tau_{e,l})]} \tag{1A}$$

$$R_l = \frac{\omega_l(1-g_l)\sinh(K\tau_{e,l})}{[2K - \omega_l(1-g_l)\sinh(K\tau_{e,l}) + 2K\cosh(K\tau_{e,l})]} \tag{1B}$$

Here, $\omega_l$, $g_l$ and $\tau_{e,l}$ denote the single scattering albedo (SSA), asymmetry parameter and extinction optical depth, respectively, of the composite layer. The parameter K is defined as:

$$K = \sqrt{(1 - \omega_l)(1 - g_l \omega_l)} \tag{2}$$

Arnott et al. (2005) used the above model to derive the form for an approximate correction factor for the aethalometer. The aethalometer uses optically-thick quartz fiber filters, which are strongly multiple scattering, transmitting only ~10% of light in the visible wavelengths. A mathematical consequence of strong multiple scattering is that the term $K\tau_{e,l}$ is much greater than unity and Eq.s (1A ) and (1B) can be replaced by simplified approximations. In contrast, the Teflon filters used in this study are optically thin and constitute a weak multiple scattering medium: they transmit 70-80% of incident visible light. Therefore, the full equations for $T_l$ and $R_l$ were solved for the filter-particle system, using a range of plausible values of dimensionless aerosol optical properties: absorption optical depth ($\tau_{a,s}$) and SSA. Two other required inputs could not be measured: the penetration depth of aerosols into the filter was assumed to be 10% of the total filter thickness, and the asymmetry parameter of the aerosols was fixed at 0.6, based on the typical values reported for biomass burning emissions (Martins et al., 1998; Reid et al., 2005). A schematic representation of the two-layer system–the aerosol laden layer with properties $T_l$ and $R_l$ and a clean filter layer with properties $T_f$ and $R_f$–is shown in Fig. 1. Transmittance and reflectance ($T_s$ and $R_s$, respectively) through the filter, when light is first incident on the aerosol-laden layer(or 'sample-side') is given by (Gorbunov et al., 2002):

$$T_s = \frac{T_l T_f}{1 - R_l R_f} \tag{3A}$$

$$R_s = R_l + \frac{T_l^2 R_f}{1 - R_l R_f} \tag{3B}$$

If the light first passes through the clean filter layer, the model predicts that transmittance, $T_c$, is still given by Eq. (3A). However, filter substrates are not uniform over their depths and have visually distinguishable front and back surfaces. Therefore, measurements of $T_s$ and $T_c$ are expected to differ. For the model substrate, reflectance $R_c$ is given by:

$$R_c = R_f + \frac{T_f^2 R_l}{1 - R_l R_f} \tag{4}$$

Attenuation (*ATN*) due to the aerosol deposit is calculated by applying Beer-Lambert's law relating to the reduction in transmittance of an exposed filter ($T_s$) relative to a blank ($T_b$) (Bond et al., 1999; Campbell et al., 1995). For a non-absorbing filter substrate, all attenuation of incident light must be caused by aerosol light absorption. Therefore, *ATN* is a measure of $\tau_{a,s}$.

$$ATN = \ln\left(\frac{T_b}{T_s}\right) = \ln\left(\frac{1 - R_b}{T_s}\right) \tag{5}$$

Here, $R_b$ is the reflectance of the blank. An alternate measure of filter-aerosol optical depth utilizes transmittance and reflectance of the clean face of the filter (Campbell et al., 1995; White et al., 2016). This reflectance measurement $R_c$ can be assumed to be approximately equal to $R_b$. PTFE blanks are non-absorbing, therefore the numerator in Eq. (5) can be replaced

by $T_b = 1 - R_b \approx 1 - R_c$. It should be noted that for translucent Teflon filters, $R_b$ and $R_c$ cannot be assumed to be exactly equal (Campbell et al., 1995; Clarke, 1982). Therefore, we represent this measure of optical depth by a separate variable, $OD_c$, where the subscript denotes that the transmittance and reflectance values used corresponds to the clean side of the filter:

$$OD_c = \ln(\frac{1-R_c}{Tc}) \qquad\qquad\qquad\qquad\qquad\qquad\qquad\qquad (6)$$

Finally, we define an optical depth measure using sample-side transmittance and reflectance, which can be interpreted as a measure of transmission of the fraction of incident radiation that is not backscattered by the filter-aerosol system:

$$OD_s = \ln(\frac{1-R_s}{T_s}) \qquad\qquad\qquad\qquad\qquad\qquad\qquad\qquad (7)$$

Values of $ATN$, $OD_c$ and $OD_s$ for a range of $\tau_{a,s}$ values (0-1) are shown in Fig. 2 for two cases: (1) highly absorbing aerosols
(SSA=0.3) and (2) highly scattering aerosols (SSA=0.95). We illustrate that $OD_c$ is nearly equal to $ATN$ for the absorbing aerosol case, but there are significant differences between the two optical measures when the filter is loaded with highly scattering aerosols. This is because a translucent substrate with a reflective coating on its back behaves like a mirror: reflectance of the substrate increases when such a coating is applied. We also find that $ATN$ shows the largest variation with SSA for a given value of $\tau_{a,s}$ while $OD_s$ exhibits the smallest variation. This can be attributed to the changing relationships
between $R_s$ and filter loading for different SSA values (see Supplement Fig. S2). For large SSA, $R_s > R_b$ and therefore, from Eq.s (5) and (7), $OD_s < ATN$. The converse is true for small SSA values. It should be noted that fixed blank optics–based on the mean of transmittance and reflectance measurements (measurement techniques are described in the following section) on 20 blank filters–were used to model $ATN$, $OD_c$ and $OD_s$. The purpose of this exercise was to illustrate the sensitivity of the above optical depth measures to SSA values of aerosols deposited on identical media.

The variation in filter optical measures with SSA was quantified by calculating the means and standard deviations of $ATN$, $OD_c$ and $OD_s$, over SSA values ranging 0.2-0.99 for each input value of $\tau_{a,s}$. A total of 500 linearly spaced points along the SSA range were used. Blank filter properties were also varied within the model in accordance with the range observed over the 20 lab blanks. For every model sample, defined by a given SSA and $\tau_{a,s}$ combination, a model blank was generated assuming a normal distribution of blank transmittance values (mean=0.7, standard deviation = 0.02). Fig. 3 shows the ratio
of standard deviation to the corresponding average values of each filter optical depth measure. The relatively low standard deviation in $OD_s$ (for most loading values) implies that this variable is a good candidate for estimating aerosol light absorption from filter optical measurements, for a wide range of aerosol types. The increase in standard deviation with increasing loading is mainly contributed by very high SSA points which are typically associated with lower absorption per unit mass: therefore very high mass loadings of such aerosols would be required to yield the upper range of the $\tau_{a,s}$. For
SSA<0.9, modeled attenuation values show little spread (<15% variability around the mean) with changing SSA. A surface plot of $OD_s$ for all model data points (0.2<SSA<0.99 and 0<$\tau_{a,s}$<1) is shown in the Supplement Fig. S4.

## 2.2 Experiments

Diverse biomass fuels including wood and needles from pine, fir and sage trees, grass, peat and cattle dung were burned in a 21 $m^3$ stainless steel combustion chamber located at Washington University (Sumlin et al. (2018); Sumlin et al. (2017)). Flaming, smoldering and mixed combustion phases were employed (see Supplement, Text S1) to generate a range of intrinsic aerosol properties: SSA values at 375, 405 and 532 nm ranged 0.4-0.99 and Absorption Ångström Exponents (AÅE) for 375-532 nm ranged 1.2-6.8. A kerosene lamp was used to generate soot particles, with an SSA of ~0.3 and AÅE within 0.70-1.1. A schematic of the experimental setup is shown in Fig. 4. Experimental conditions and intrinsic aerosol optical properties for each fuel-combustion phase combination are listed in Table 1. Approximately 10-50 g of a given type of woody biomass/grass/dung was placed in a stainless-steel pan and ignited using a flame. It was either allowed to continue flaming or brought to a smoldering phase by starving the flame with a lid. In the same type of pan, 5-15 g of peat was smoldered by using a ring heater to raise its temperature to 200□C. In one set of experiments, smoke from the chamber was directly sampled, while in another set, a hood placed over the pan was used for sampling the aerosols. The chamber exhaust was closed during the burns. The outlet from the hood or chamber was passed through a diffusion dryer and a semi-volatile organic compound (SVOC) denuder into a mixing volume, from which aerosols were continuously sampled by the four IPNs.

During each burn, optical (absorption and scattering coefficients) signals were monitored using IPNs until a steady state was reached. During the steady state, particle samples were collected on 47 mm PTFE membrane (Pall) filters. The filter sampling flow rate was set to 5 liters per minute and the sampling durations were between 2 and 20 minutes. For each filter sample, $\tau_{a,s}$ of the deposited aerosols was calculated from the absorption coefficients measured using the IPNs:

$$\tau_{a,s} = \frac{b_{abs,av} \times Q \times t_s}{10^9 A_s} \qquad (8)$$

where $b_{abs,av}$ is the average absorption coefficient (in $Mm^{-1}$) during the sampling duration $t_s$ (in min), Q is the flow rate (in liters per minute or lpm) through the filter and $A_s$ is the filter sample area (in $m^2$). Optical depth $\tau_{a,s}$ for the samples in this study ranged between 0.01 and 0.68. The uncertainty in these estimates was predominantly from the standard deviation in $b_{abs,av}$ over the averaging interval, and was within 10% for all samples. Values of $b_{abs,av}$ at 532 nm ranged from ~300 $Mm^{-1}$ for smoldering samples to ~20000 $Mm^{-1}$ for flaming phase samples; the corresponding range at 375 nm was ~3000-30000 $Mm^{-1}$.

Sample-side transmittance ($T_s$) and reflectance ($R_s$) for the filter samples were measured using a Perkin-Elmer LAMBDA 35 UV-vis spectrophotometer (described in Zhong and Jang (2011)). This instrument contains an integrating sphere and two sample holders. Transmittance was measured by placing the sample in the first holder ahead of the sphere, in the direction of the sample beam, while a white standard was placed in the second holder (behind the sphere). Reflectance was measured by keeping the first holder empty and placing the sample in the second holder. Both measurements were performed on the sample face of the filter: light was incident on the side that was exposed to the sample air. Each measurement was

normalized to the baseline transmittance/reflectance value of the measurement system: between every 10 sample scans, transmittance/reflectance were measured with no sample placed in the first holder and a white standard was placed in the second holder. Sample transmittance/reflectance values were then divided by the corresponding baseline. Only $T_s$ and $R_s$ were measured for all samples in this study as model results indicated that $OD_s$ is better suited than $OD_c$ for estimating $\tau_{a,s}$ (Fig. 3). To test the validity of this assumption, transmittance and reflectance were also measured on the clean side of the filter ($T_c$ and $R_c$, respectively) for a subset of the samples (n=54). This subset corresponded to samples collected during 17 biomass burning experiments which yielded aerosols with SSA (375, 405 and 532 nm) ranging 0.54-0.99. For all samples, we found $T_s > T_c$.

From normalized $T_s$ and $R_s$ measurements, $OD_s$ was calculated using Eq. (7). When this equation is applied to blank filters, it results in $OD_s$ values between 0.01-0.03. A wavelength dependent "blank optical depth" was subtracted from the sample data. Triplicate transmission and reflection measurements were used to estimate measurement uncertainty, which is attributable to random fluctuations in the measurements. Means and standard deviations of the $OD_s$ values calculated from the replicate measurements yielded an uncertainty (ratio of standard deviation to mean) of 5% in $OD_s$. Similarly, $OD_c$ was calculated for the 54-sample subset.

A correction factor ($C$) that captures the net effect of multiple scattering and aerosol loading can be defined as:

$$\tau_{a,s} = C \times OD_s \implies C = \frac{\tau_{a,s}}{OD_s} \tag{9}$$

## 3 Results and discussion

The model described in Section 2.1 was used to calculate filter optical depths for each experimental sample (using measured $\tau_{a,s}$ and SSA values as inputs). Modeled and experimental values of $OD_s$ for the samples are shown in Fig. 5. The two datasets are highly correlated (Pearson $R$= 0.92), but the model predicted larger values of $OD_s$ than those experimentally determined. This disagreement may partially be due to differences between assumed parameters in our model and their real-world values. It is also possible that assuming an average propagation direction of diffuse radiation within the two-stream approximation (Arnott et al., 2005; Sagan and Pollack, 1967) causes this systematic difference.

In Fig. 6, we combined all experimental ($\tau_{a,s}$ versus $OD_s$) data corresponding to the three wavelengths since our measurements showed no clear stratification with varying wavelength. The relationships between $\tau_{a,s}$ and modeled values of $OD_s$, $OD_c$ and $ATN$ are presented in the Supplement (Fig. S5); modeling predicts the lowest scatter in the $\tau_{a,s}$-$OD_s$ curve (Fig. S5). Further, Fig. 7 shows $\tau_{a,s}$ plotted against measured $OD_s$ and $OD_c$ (at all three wavelengths) for the aforementioned 54 filter sample subset, demonstrating that $\tau_{a,s}$ is better correlated with $OD_s$ than with $OD_c$. Ordinary least-squares regression was applied to obtain power-law fits included in the plot legend. The corresponding relationship for all points in Fig. 6 is given by ($R^2$ = 0.87):

:

$$\tau_{a,s} = 0.48 \, (OD_s)^{1.32} \tag{10}$$

Also shown in the figure are estimated $\tau_{a,s}$ using a constant correction factor $C$ of 0.67 proposed by Zhong and Jang (2011) (black perforated line); this correction factor clearly overestimates $\tau_{a,s}$ for most $OD_s$ values investigated in this study. We find our data to be better represented by an approximate $C = 0.46$ based on a linear least-squares fit ($R^2 = 0.79$). However, any constant $C$ value does not capture the non-linearity of the interaction between aerosol properties and the multiple-scattering within the filter medium. It should be noted that $C$ in Eq. (9) represents the net effects of all filter artifacts. There are measurement errors associated with both $OD_s$ and $\tau_{a,s}$, and therefore, $C$ contains propagation of uncertainties from both parameters. There was no correlation between $C$ and $OD_s$ (see Fig. S6). We observed an inverse relationship between $C$ and SSA (Fig. 8), consistent with results the from the two-stream radiative transfer model. For a given value of $\tau_{a,s}$, $OD_s$ will always be higher for aerosols with higher SSA values. Consequently, we should expect $C$ to decrease with increasing SSA; this decreasing relationship in our data is given by:

$$C = -0.76 * SSA + 1.02 \tag{11}$$

Values of $C$ and SSA for individual samples (shown in supplemental Fig. S6B) were aggregated into five SSA bins to demonstrate the inapplicability of an empirical correction factor formulation to low SSA data points in this study. The large spread in $C$ values for low SSA is likely due to noise amplification from dividing two small ($\tau_{a,s}$ and $OD_s < 0.2$) numbers. For SSA>0.6, the above linear fit holds.

**5 Summary**

We evaluated the relationship between *in-situ* aerosol light absorption and attenuation of aerosol deposits on Teflon filters for combustion aerosols (encompassing $0.25 \leq SSA \leq 0.99$), at 375, 405 and 532 nm wavelengths. An empirical non-linear relationship was found between the absorption optical depth of sampled aerosols and attenuation through filter samples; the nature of this function was consistent with predictions from a two-stream radiative transfer model of the filter-aerosol system. Following Eq. (10), we propose the estimation of aerosol MAC ($m^2g^{-1}$) values from filter $OD_s$ measurements using:

$$MAC = [0.48 \, (OD_s)^{1.32}] \frac{A_s}{m} \tag{12}$$

where $A_s$ is the filter sample area (in $m^2$) and $m$ is the mass on deposited particles (in g). Additionally, aerosol absorption coefficients ($b_{abs}$; in $Mm^{-1}$) can also be calculated using:

$$b_{abs} = [0.48 \, (OD_s)^{1.32}] \frac{10^9 A_s}{Q \times t_s} \tag{13}$$

The quantities $Q$ and $t_s$ are as used in Eq. (8). Caution must be taken, as suggested by the two-stream model results, on the limits of applicability of the empirical relationships (equations 10-13)–significant errors could result from application of the relationships if the aerosol SSA>0.9 and $OD_s$ values are beyond the range of this work.

Teflon filters are routinely used for gravimetric and elemental analysis across monitoring networks (Chow et al., 2010; Snider et al., 2015; Solomon et al., 2014), as well as field and laboratory source characterization studies. In many measurement systems, such as the Hybrid Integrating Plate and Sphere (HIPS) method (Bond et al., 1999) used by the IMPROVE network, transmittance and reflectance are measured on the clean side of the filter and the optical depth $OD_c$ is calculated (Campbell et al., 1995; White et al., 2016). The relationship between aerosol optical depth, $\tau_{a,s}$, and $OD_c$ showed a larger variability across varying SSA than that between $\tau_{a,s}$ and $OD_s$. Therefore, we suggest further evaluation of $OD_s$ as an optical depth measure that can be empirically connected to particulate phase light absorption for a range of aerosol types.

## Supplement

Includes a schematic of the experimental setup  (Text S1), detailed equations for the two-stream model (Text S2, Figure S1), supplemental modeling results (Text S2, Figures S2-S5), and correction factor plots (Text S3, Figure S6).

## Acknowledgements

This work was partially supported by the National Science Foundation under Grant No. AGS1455215, NASA ROSES under Grant No. NNX15AI66G, and the International Center for Energy, Environment and Sustainability (InCEES) at Washington University in St. Louis.

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

Table 1: Number of burns conducted, and filter samples collected for each fuel type and combustion phase in this study. Intrinsic optical properties of emissions from each study condition are also given.

| Fuel | Combustion phase | SSA | | | AÅE | Number of burns | Number of filter samples |
|------|------------------|--------|--------|--------|-----------|-----------------|--------------------------|
| | | 375 nm | 405 nm | 532 nm | 375-532 nm | | |
| Dung | smoldering | 0.86 | 0.95 | 0.98 | 6.1-6.6 | 6 | 7 |
| Peat | smoldering | 0.92 | 0.97 | 0.99 | 4.8-6.8 | 2 | 3 |
| Sage | smoldering | 0.75-0.87 | 0.86-0.93 | 0.93-0.97 | 2.8-5.3 | 4 | 14 |
| | mixed | 0.56-0.77 | 0.69-0.84 | 0.71-0.87 | 1.5-2.3 | 3 | 7 |
| | flaming | 0.43-0.65 | 0.62-0.69 | 0.69-0.77 | 0.9-1.4 | 2 | 5 |
| Grass | smoldering | 0.74 | 0.87 | 0.94 | 3.2-4.7 | 3 | 7 |
| | flaming | 0.76 | 0.81 | 0.85 | 1.7 | 1 | 3 |
| Lodgepole pine | smoldering | 0.84 | 0.93 | 0.97 | 4.2 | 2 | 3 |
| Ponderosa pine | mixed | 0.61-0.84 | 0.74-0.91 | 0.76-0.95 | 1.2-3.0 | 4 | 9 |
| | flaming | 0.56 | 0.65 | 0.65 | 0.7 | 1 | 2 |
| Douglas fir | mixed | 0.82 | 0.89 | 0.93 | 2.7 | 1 | 2 |
| | flaming | 0.60 | 0.70 | 0.71 | 0.9 | 1 | 3 |
| Hardwood pellets | mixed | 0.80-0.87 | 0.92-0.95 | 0.95-0.98 | 4.1-6.1 | 1 | 3 |
| Kerosene | flaming | 0.27 | 0.30 | 0.31 | 0.7-1.1 | 3 | 7 |

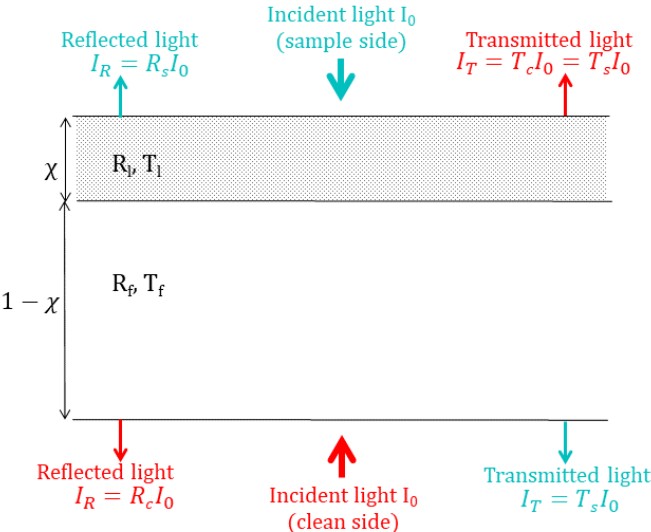

**Figure 1: Two-layer model of a filter sample consisting of an aerosol laden layer 'l' and a clean layer 'f'.**

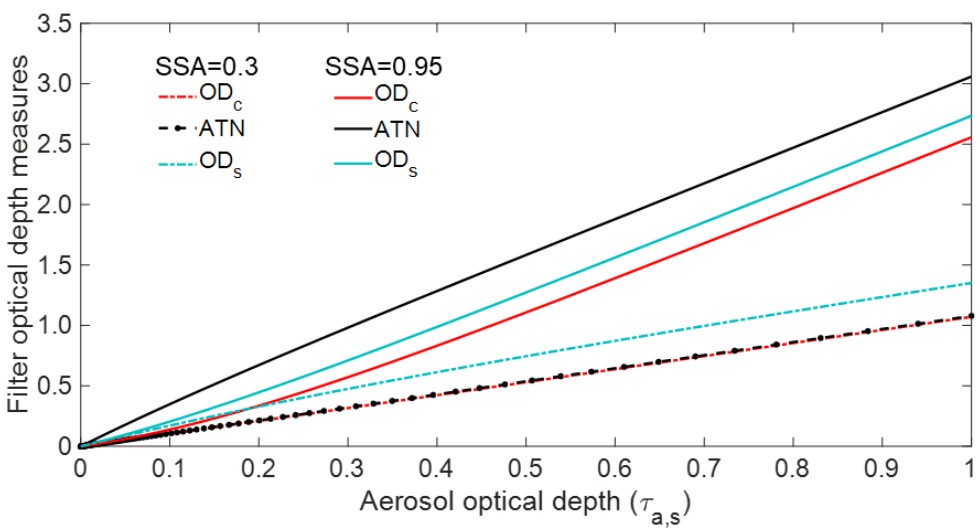

**Figure 2: Modeled values of filter optical depth measures ($OD_c$, $ATN$ and $OD_s$) with increasing aerosol optical depth ($\tau_{a,s}$) of deposited highly absorbing (SSA=0.3) or highly scattering (SSA=0.95) aerosols. Fixed blank optics were assumed.**

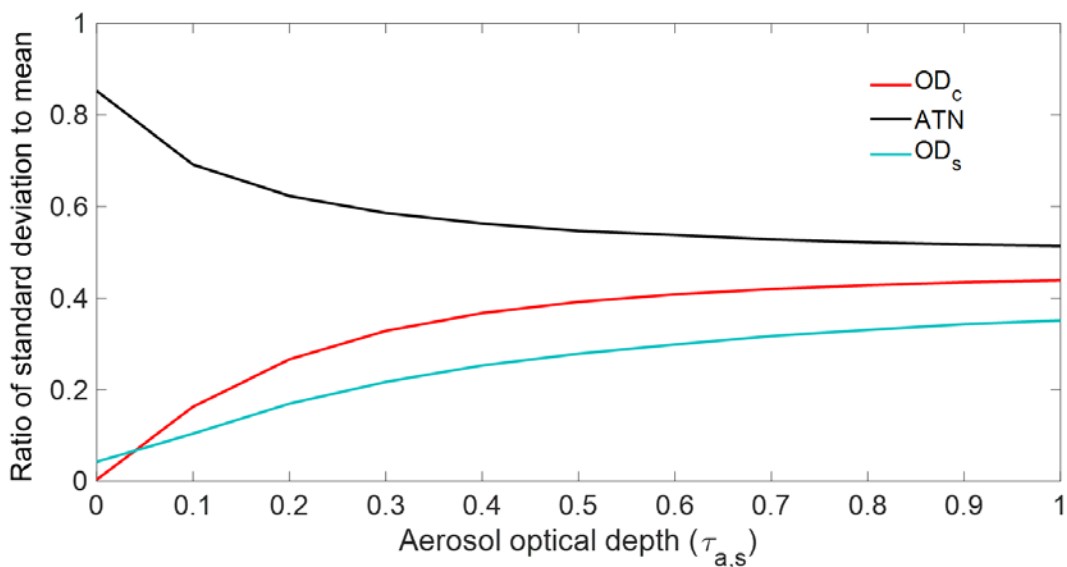

**Figure 3: Ratio of standard deviations to means of modeled filter optical depth measures ($OD_c$, $ATN$ and $OD_s$) averaged over 500 equally spaced SSA points ranging 0.2-0.99, corresponding to each value of aerosol optical depth ($\tau_{a,s}$). Blank optics were randomly generated for each sample point from a normal distribution with mean=0.7 and standard deviation=0.02.**

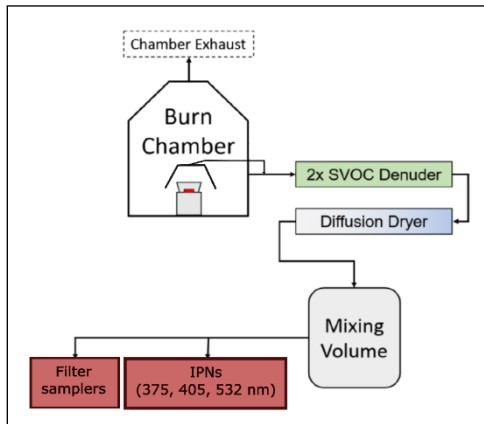

**Figure 4: Schematic representation of the experimental setup. Inlet to the semi-volatile organic compound denuder was taken from either the chamber sampling port or the hood. IPN stands for integrated photoacoustic-nephelometer spectrometers.**

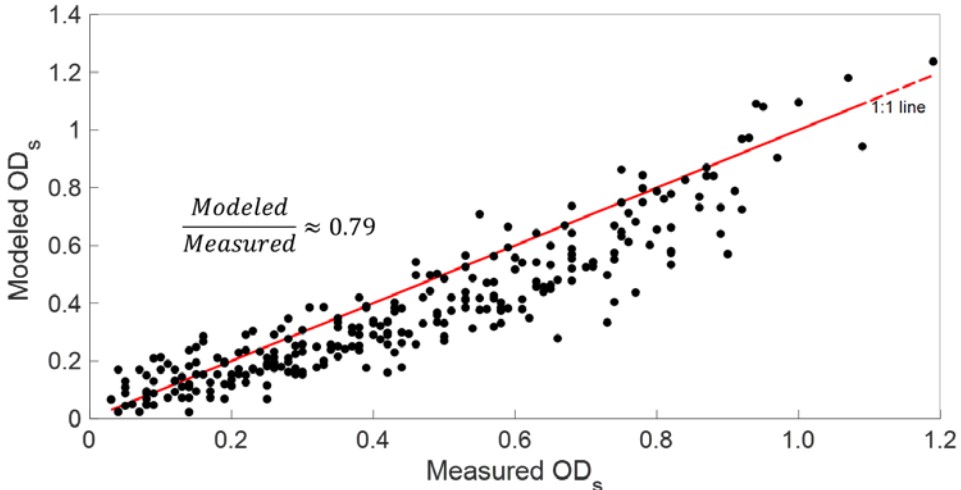

**Figure 5: Modeled filter optical depth ($OD_s$) for absorption optical depth and single scattering albedo values of the aerosols sampled in this study compared with the corresponding filter measurements. A 1:1 line is shown in red. The average ratio of modeled to measured $OD_s$ is 0.79.**

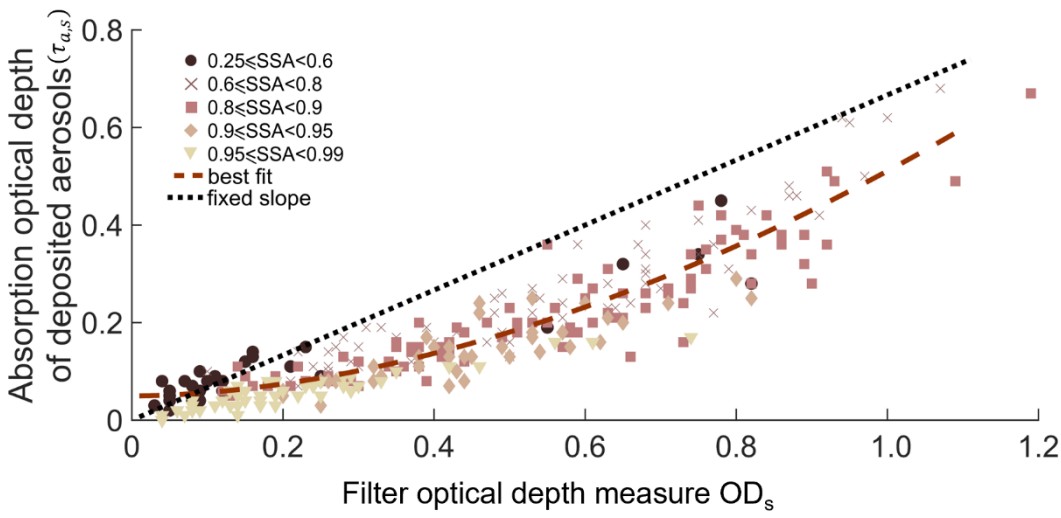

**Figure 6: Relationship between *in-situ* aerosol optical depth (τ_{a,s}) and filter optical depth *OD_s* for all (n=75) samples, measured at 375, 405 and 532 nm (N=225 data points). The best fit curve is given by Eq. (10), with R² = 0.87. The black perforated line has a fixed slope of 0.67 per Zhong and Jang (2011). Uncertainties (1 standard deviation) in *OD_s* ranged 2-5%, while those in τ_{a,s} were 5-10%.**

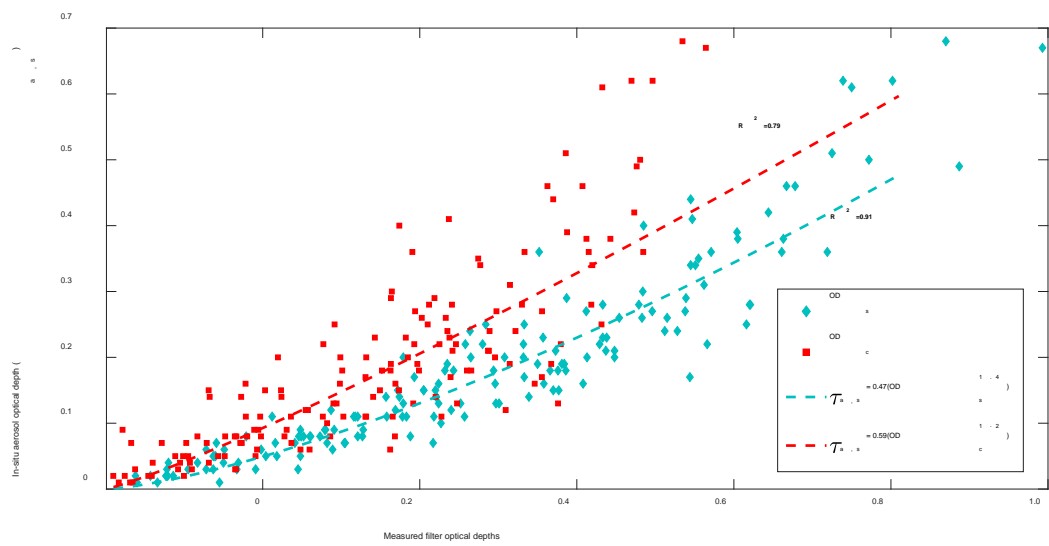

**Figure 7: Relationship between *in-situ* aerosol optical depth (τ_{a,s}) and measured values of filter optical depth measures *OD_c* and *OD_s* for a subset of 54 filter samples, measured at 375, 405 and 532 nm (N=162 data points). Uncertainties were as in Fig. 6.**

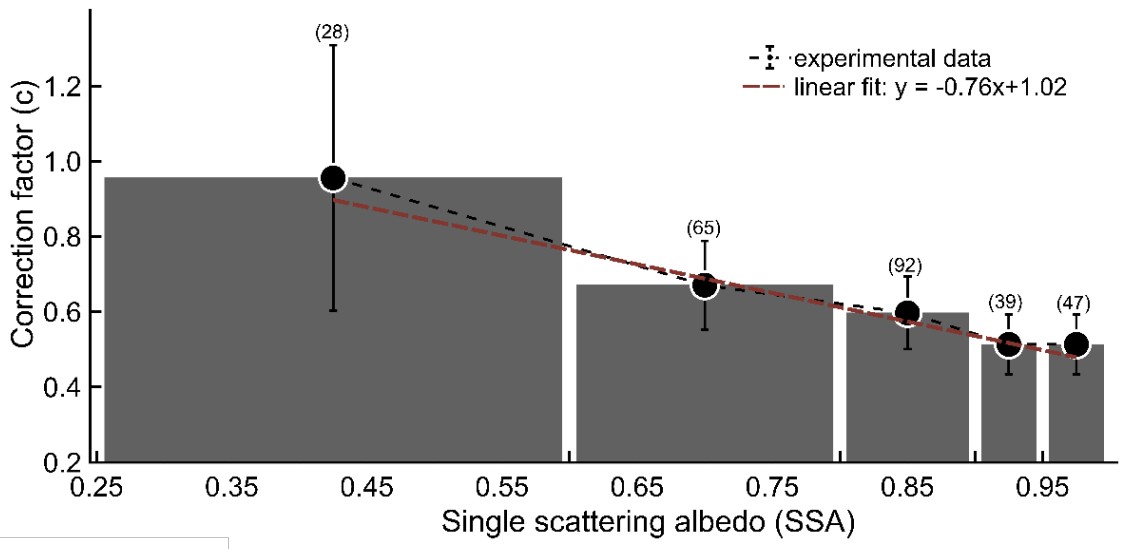

**Figure 8: Correction factor C for filter artifacts as a function of single scattering albedo of the deposited aerosols. Error bars show one standard deviation around the mean. The numbers in parentheses denote the number of data points in each bin.**