# Peer review of "Aerosol light absorption from optical measurements of PTFEmembrane filter samples: sensitivity analysis of optical depth measures"

_Atmospheric Measurement Techniques, 2018_

## Referee Comment (RC1) · W. White (Referee) · 29 Aug 2018

**GENERAL COMMENTS**

This discussion paper compares optical measurements of PTFE-membrane filter samples with in situ measurements of the aerosols from which they were collected. It is novel in its approach and inspiring in its ambition. The authors generated carbonaceous aerosols with a range of optical properties from the combustion of biomass fuels and kerosene, using an integrated experimental system to collect paired in-situ and

filter-based optical data. A virtual AOD (absorption optical depth) was calculated for each sample, as the product tau of the in-situ photoacoustic absorption coefficient and the length of the sampled air column. The sample's actual AOD was independently estimated from direct measurements of overall filter reflectance and transmittance, presumably with a single or double integrating sphere system that is not described. The authors summarize the empirical relationship between the two absorption measures with a "best fit" formula they motivate as "established in conjunction with" predictions from a two-stream radiative transfer model.

It feels awkward for me to comment as a referee on the titular "implications for particulate matter monitoring networks", which is based on a "case study" of the IMPROVE haze monitoring network. Having coauthored a recent paper (White et al., 2016) validating the filter-absorption measurement reported by IMPROVE, I may be viewed as conflicted when evaluating this discussion paper's interpretation of those data. It seems fair, however, to observe that no one should expect the absorption characteristics of an agglomerated deposit of particulate matter to be exactly those of the morphologically complex and often fragile aerosol particles from which it was filtered. Figures 2 and 3 indeed show a great deal of scatter in the relationship between these two measurement types, even when obtained within the same laboratory system by the same investigators. The discussion paper undertakes to produce an empirical calibration for a specific filter-based measurement in terms of in-situ aerosol measurements. Its finding, that applying the same calibration to a completely different measurement system in Figure 5 produces an unsatisfactory relationship, should not surprise anyone who recalls the authors' earlier comment (page 2, line 16): "Typically, correction algorithms . . . are specific to a given measurement system." Moreover it is misleading to call this foreign calibration of IMPROVE data a correction: the designation of IMPROVE data as Fabs rather than babs is explicitly "intended to remind users of {their} origin in a filter-based measurement" (White et al., 2016), with an explicitly noted bias estimated in the range $1 \leq$ Fabs/babs $\leq 2$.
The questions and concerns raised below about the filter-based measurements at Washington University, and their relationship to the IMPROVE measurement, could be directly addressed by reanalyzing the same sample filters on the HIPS system used at UC Davis for the IMPROVE measurements. The ability to revisit previously analyzed samples is an important strength of filter-based absorption methods, and both laboratories could learn by taking advantage of this opportunity. A comparison between the WUSTL in-situ measurements and UCD HIPS results may have only limited relevance for the predominantly rural and remote IMPROVE network, which among other differences samples generally mixed and well-aged emissions rather than strictly fresh fumes and smokes. The HIPS group, with whom I am in contact, would nevertheless be delighted to collaborate in exploring the possibilities.

SPECIFIC COMMENTS

1. The entire description of the authors' filter optics measurements is a scant two sentences: "Transmittance (T) and reflectance (R) for the filter samples were measured using a Perkin-Elmer LAMBDA 35 UV-vis spectrophotometer (described in Zhong and Jang (2011)). Attenuation (ATN) through the filter samples was calculated using (Bond et al., 1999; Campbell et al., 1995): ATN = ln((1-R)/T)". This hardly specifies what was measured, as detailed in the next two questions below.

(a) Which "R" are we talking about here? Is it the reflectance of the "dirty" side of the filter, as in the old British Smoke Shade (BSS) method? This is what the 2011 Zhong and Jang reference indicates in its Figure 1b. Or is it the reflectance of the "clean" side, as specified in the 1995 Campbell et al. paper ("All measurements were made with the side of the filters containing the particle deposit facing away from the laser beam") and continued in all subsequent IMPROVE measurements (White et al., 2016)? For a moderately loaded sample filter, the difference is apparent to the naked eye. Moreover, light reflected from the "dirty" side depends nonlinearly on the AOD: the portion reflected from the pristine layer of substrate must pass twice through the deposit layer, creating a quadratic dependence on the particle loading (cf. Equation

4, Gorbunov et al., 2002). The fact that this quadratic dependence appears in the discussion paper's radiative transfer model (Equation 6B) implies R was measured on the sampled side of the filter.

(b) What light-collecting geometry was used to measure R and T? On the one hand Zhong and Jang (2011) describe separate measurements of T and R, with a single integrating sphere accessory operated in two different configurations. This resembles the LISA (Laser Integrating Sphere Analysis) described by Campbell et al. (1995). On the other hand the 2016 paper by Pandey, Pervez, and Chakrabarty, which seems to be a precursor to the paper under review, describes using a PerkinElmer LAMBDA 35 – the very same instrument? – with a double integrating sphere system to make the same kind of measurements. These two arrangements require different approaches to interpreting and calibrating the raw data: jointly measured T and R signals from a pair of integrating spheres (e.g. Pickering et al., 1992), or from the hybrid of integrating sphere and plate used by IMPROVE (White et al., 2016), are coupled in ways that successive measurements of the individual signals are not.

2. Light attenuation through a filter sample is generally defined as $ATN = \ln(T_b/T_s)$, where $T_b$ is the transmittance of the unsampled filter blank and $T_s$ is that of the same filter after sampling (e.g. Pandey et al., 2016). The alternative measure $\ln((1-R)/T)$ introduced by Campbell et al. (1995) uses the reflectance $R_c$ of the sampled filter's "clean" (unexposed) side as an estimate $R_c \approx R_b$ for the reflectance of the unexposed blank. The expectation that bare PTFE does not absorb, so that $T_b = 1-R_b \approx 1-R_c$, then justifies the substitution of $\ln((1-R_c)/T_s)$ for attenuation as usually defined. It should be noted that Campbell et al. (1995), Bond et al. (1999), and White et al. (2016) all refer to $\ln((1-R/T)$ as a measure of AOD rather than attenuation. I think this discussion paper's unannounced and unmotivated change in terminology is likely to introduce confusion in the majority of readers who are not familiar with the rationale just outlined. And again, this rationale strictly requires that reflectance be measured for $\ln((1-R/T)$ on the "clean" $R = R_c$ side of the sampled filter; it's not clear what relevance the quantity $\ln((1-R)/T)$

would have to attenuation or any other interesting sample property if R were measured on the side darkened by the collected sample deposit.

3. The experimental program described in this discussion paper was well designed to test our standard accounts of radiative transfer in filter samples, providing real data for a closure study that I have not seen attempted before. The authors have produced 75 experimental filters according to Table S1, presumably running each through their spectrophotometer system before sample collection to characterize the substantial variability in individual PTFE membranes that was highlighted by White et al. (2016). They have modeled radiative transfer through the filter at an appropriate level of detail, and have measured relevant optical properties of both the aerosols and collected particulate matter. They have suggested plausible estimates for unmeasured parameters such as penetration depth and scattering asymmetry. The sampled aerosols were each characterized at three different wavelengths, which should yield $3 \times 75 = 225$ independent sets of model data inputs. I was expecting these substantial interwoven efforts to culminate in illuminating comparisons, presented in formats such as scatterplots of modeled vs. measured R, T, and ATN. Absent any clear link to the measurements, what was the intent of the modeling?

4. Relatedly, in what sense is Equation 7 a "best fit" to the observed relationship between ATN and AOD? What does it mean to say "the nature of this function was consistent with predictions from a two-stream radiative transfer model"? Is Equation 7 simply an OLS regression of logarithms, or is it somehow informed by results from the radiative transfer modeling? What sort of functional form does the two-stream model predict when it is run over a wide range of sample loadings from a representative aerosol of fixed optical characteristics?

5. The only results I see explicitly attributed to the two-stream model are the colored wedge shapes in Figure 2. I don't understand why these don't show y-axis ATN declining toward zero, the model prediction for a blank, as the virtual absorption optical depth of the aerosol sample approaches zero on the x-axis.

6. The two-stream radiative transfer modeling framework presented here, along with its hyperbolic solutions, is widely known also as Kubelka-Munk theory, after an extensive older literature arising in paper and paint research (e.g. Kortum, 1969). Referencing this connection in passing might attract a few additional readers.

TECHNICAL CORRECTIONS

Line 25, page 6 of main paper: The reference to Figure 1 appears to be in error.

Section S4 of supplement: (a) In the IMPROVE network, the equivalent length of the sampled air column can vary somewhat from sample to sample. The 93 km value is a design target, subject to limited and occasional flow and timing variations, and is not "fixed". (b) The IMPROVE data warehouse at CIRA holds Fabs and EC values from 2010 for 172 sites; in no year has the network ever operated 223 sites. (c) Authors who use IMPROVE data are asked, as a favor to aid in tracking these papers, to include the seminal 1994 paper by Malm et al. as a reference.

REVIEW REFERENCES NOT ALREADY INCLUDED IN THE DISCUSSION PAPER

G. Kortum (1969) Reflectance Spectroscopy: Principles, Methods, Applications. Translated from the German by J.E. Lohr, Springer-Verlag New York.

W. C. Malm, J. F. Sisler, D. Huffman, R. A. Eldred, and T. A. Cahill (1994), Spatial and seasonal trends in particle concentration and optical extinction in the United States, J. Geophys. Res., 99, 1347-1370.

J.W. Pickering, C.J.M. Moes, H.J.C.M. Sterenborg, S.A. Prahl, and M.J.C. van Gemert (1992) Two integrating spheres with an intervening scattering sample. J. Opt. Soc. Am. A, v9, pp. 621-631

ACKNOWLEDGMENT Scott Copeland, a coauthor of the original Campbell et al. (1995) paper, helped focus my thoughts by noting the important distinction between calibration and correction.

---

## Referee Comment (RC2) · Anonymous Referee #2 · 8 Oct 2018

General comments:

In this manuscript the authors investigated the relationship between absorption optical depth of aerosol carbonaceous particles derived from in situ measurements and attenuation obtained using filter based method. The authors derived empirically an estimation of the aerosol mass absorption coefficient (MAC) from PTFE filter measurements, and they applied this formulation on an independent dataset from IMPROVE. They noticed that this correction resulted in a significant change in the attenuation for low aerosol concentrations. The introduction of the paper is well written, but the au-

thors have not clearly reported their results in the abstract. I also suggested that the authors include in the main text some of the information that is presented in the supplement, specifically sessions S1 and S4, as in my opinion those sessions are very relevant for the understanding of this study. Also, although the authors have included a reference about the filter based method applied, there is no other indication on how the reflectance and transmittance measurements were taken. At a minimum, the authors could report if those measurement are integrated quantities or if they were taken at a given angle. Finally, I think the author could discuss and explore the results found in the case study in more details.

Specific Comments:

P1L15: When you use the term "light absorption", do you mean the mass absorption coefficient or absorption optical depth or the linear absorption coefficient? The same thing for "attenuation measurements". Are you referring to extinction, absorption or something else? If something else, please define.

P1L18: In the sentence "we find the ratio between in situ absorption and bulk attenuation to be inversely proportional to aerosol single scattering albedo", please clarify the term "absorption". Are you referring to absorption optical depth?

P1L21: What are the results from the case study? You have not mentioned that in the abstract.

P1L27: For completeness, you could add here a few examples of known carbonaceous sources with different MAC.

P2L18: Another relevant reference here: Martins, J. V., Artaxo, P., Kaufman, Y. J., Castanho, A. D., and Remer, L.: Spectral absorption properties of aerosol particles from 350–2500nm, Geophys. Res. Lett., 36, L13810, https://doi.org/10.1029/2009GL037435, 2009.

P3L26: You mentioned here "sampled by the four IPNs" but I don't see any reference to

these measurements later in the text. It might be interesting to comment if the average absorption coefficients was derived using those four instruments. Also, it would be interesting if you could report/discuss for each sample type.

P4L5: I suggest adding here a figure or table reporting the average values of the absorption coefficient in Mm-1 as well as the standard deviation for the different samples. Also, I would move table S1 that is currently in the abstract to the main manuscript.

P5L5: In the sentence: "the penetration of aerosols into the filter was assumed to be 10%". Could you please clarify the meaning of that? Do you mean 10% of the mass collected? How do you estimated this number? Figure 5 shows concentrations in the range from 0.01 to near 10ugm-3. Would you expect the percentage be strongly dependent of the mass concentration collected on the filter?

P5L13: I think it is missing here an evaluation of the uncertainties in the calculation of ATN.

P6L24: I also suggest moving session and figure S4 to the main manuscript as that is important for the understanding of Figure 5.

P7L9: How these method compare/differ from the derivation of the mass absorption efficiency presented in (Martins et al., 2009)?

Figures 2, 3 and 5: These plots are missing error bars. You should also add information about the uncertainty of the measurements in the legend of each figure.

---

## Author Comment (AC1) · 3 Nov 2018

We have addressed the reviewers' comments below and made substantial changes to the manuscript, including the title and abstract. We have removed the case study–application of our empirical findings to IMPROVE network data–because of crucial differences between their measurement method and ours. We have also elaborated on how the radiative transfer modeling informed this study.

Reviewer 1 General comments of the "implications" discussed in our manuscript. "The

discussion paper undertakes to produce an empirical calibration for a specific filter-based measurement in terms of in-situ aerosol measurements. Its finding, that applying the same calibration to a completely different measurement system in Figure 5 produces an unsatisfactory relationship, should not surprise anyone who recalls the authors' earlier comment"

We are grateful to Dr. White for his insightful, detailed comments on our manuscript. We found these comments very helpful in refining the thought process behind this paper, leading to a substantial revision. We agree that the empirical equation developed in this work is not applicable to IMPROVE data and have entirely removed the "implications" section. We have also highlighted how the filter optical measurements used here are distinct from those in literature and provided our reasoning for defining this alternate measure of optical depth. Further, several new figures have been added to the manuscript and the supplement to better illuminate our modeling results (Figures 2, 3, 5, S2-S5).

Specific comments 1 and 2. "The entire description of the authors' filter optics measurements is a scant two sentences: "Transmittance (T) and reflectance (R) for the filter samples were measured using a Perkin-Elmer LAMBDA 35 UV-vis spectrophotometer...... if R were measured on the side darkened by the collected sample deposit."

The filter optical measurements are now described on page 6, Line 5. The rationale behind using reflectance measurements on the "sample" side of the filter is detailed in Section 2.1. Briefly, we recognize that the quantity $\ln((1-R_s)/T_s)$, where $R_s$ and $T_s$ denote reflectance and transmittance measurements made on the sample side, is not equivalent to Beer-Lambert attenuation. We now explicitly define it in the manuscript as a separate measure of filter optical depth (represented by ODs). We experimentally observed that this alternate filter optical measure is remarkably well-correlated with in-situ absorption optical depth. We also found that the two-layer filter model supported our findings (Fig.s 2 and 3 in the main manuscript) and indicated that the reason for

the above constrained relationship is the behavior of Rs as a function of filter loading, for different values of SSA (Fig. S2 in the Supplement).

Specific comments 3 and 4. "The experimental program described in this discussion paper was well designed to test our standard accounts of radiative transfer in filter samples, providing real data for a closure study that I have not seen attempted before. . . . Absent any clear link to the measurements, what was the intent of the modeling? Relatedly, in what sense is Equation 7 a "best fit" to the observed relationship between ATN and AOD?....What sort of functional form does the two-stream model predict when it is run over a wide range of sample loadings from a representative aerosol of fixed optical characteristics?"

A comparison between modeled and measured values of ODs is now shown in the manuscript (Fig. 5). The predicted relationship between AOD and ODs (along with other commonly used filter optical measurements) is demonstrated in the Supplement (Fig. S5) and discussed in the manuscript. Due to the quadratic dependence of Rs on particle loading, a non-linear function form is required to capture the relationship between ODs and AOD for any SSA value, but the coefficients of such a relationship (say, power law) are not consistent across all SSA values.

Specific comment 5 "The only results I see explicitly attributed to the two-stream model are the colored wedge shapes in Figure 2. I don't understand why these don't show y-axis ATN declining toward zero, the model prediction for a blank, as the virtual absorption optical depth of the aerosol sample approaches zero on the x-axis."

This was due to a calculation error at zero aerosol optical depth which has now been fixed.

Specific comment 5 "The two-stream radiative transfer modeling framework presented here, along with its hyperbolic solutions, is widely known also as Kubelka-Munk theory, after an extensive older literature arising in paper and paint research (e.g. Kortum, 1969). Referencing this connection in passing might attract a few additional readers."

We have added a reference to the Kubelka-Munk theory to the manuscript.

Reviewer 2 "In this manuscript the authors investigated the relationship between absorption optical depth of aerosol carbonaceous particles derived from in situ measurements and attenuation obtained using filter based method. The authors derived empirically an estimation of the aerosol mass absorption coefficient (MAC) from PTFE filter measurements. . .. Finally, I think the author could discuss and explore the results found in the case study in more details."

We thank the reviewer for their helpful and encouraging comments. Throughout the manuscript, we have now ensured technical precision and clarity in our language. Relevant experimental details have been added and uncertainties have been stated in our figure captions. Finally, because of a crucial difference between our measurements and those in the IMPROVE network, we have removed the case study section from the manuscript.

P1L15: "When you use the term "light absorption", do you mean the mass absorption coefficient or absorption optical depth or the linear absorption coefficient? The same thing for "attenuation measurements". Are you referring to extinction, absorption or something else? If something else, please define." P1L18: In the sentence "we find the ratio between in situ absorption and bulk attenuation to be inversely proportional to aerosol single scattering albedo", please clarify the term "absorption". Are you referring to absorption optical depth?"

We have carefully gone through the manuscript to ensure that all optical quantities are defined and referred to more specifically.

P1L21: "What are the results from the case study? You have not mentioned that in the abstract."

The case study has been removed.

P1L27: "For completeness, you could add here a few examples of known carbonaceous sources with different MAC."

Typically observed ranges of carbonaceous aerosol MAC values have been added at P2L2.

P2L18: "Another relevant reference here: Martins, J. V., Artaxo, P., Kaufman, Y. J., Castanho, A. D., and Remer, L.: Spectral absorption properties of aerosol particles from 350–2500nm, Geophys. Res. Lett., 36, L13810, https://doi.org/10.1029/2009GL037435, 2009."

This reference has been added to the manuscript.

P3L26: "You mentioned here "sampled by the four IPNs" but I don't see any reference to these measurements later in the text. It might be interesting to comment if the average absorption coefficients was derived using those four instruments. Also, it would be interesting if you could report/discuss for each sample type."

A sentence on the range of absorption coefficient values has been added at page 6, line 5. Given the huge variability in absorption and scattering coefficients, we only discuss the dimensionless optical properties calculated from these measurements (aerosol optical depth and single scattering albedo).

P4L5: "I suggest adding here a figure or table reporting the average values of the absorption coefficient in Mm-1 as well as the standard deviation for the different samples. Also, I would move table S1 that is currently in the abstract to the main manuscript."

We have moved the table of experiments to the main manuscript.

P5L5: "In the sentence: "the penetration of aerosols into the filter was assumed to be 10%". Could you please clarify the meaning of that? Do you mean 10% of the mass collected? How do you estimated this number? Figure 5 shows concentrations in the range from 0.01 to near 10ugm-3. Would you expect the percentage be strongly dependent of the mass concentration collected on the filter?"

We assumed that aerosols only penetrated the first 10% of the filter thickness. Membrane filters are expected to have a low penetration depth, so we assumed a reasonable value here. We expect the general trends from our model to hold even with variations in penetration depth. Our supplement now includes a figure (Fig. S3) showing the effect of this parameter on transmittance and reflectance.

P5L13: "I think it is missing here an evaluation of the uncertainties in the calculation of ATN. Figures 2, 3 and 5: These plots are missing error bars. You should also add information about the uncertainty of the measurements in the legend of each figure."

The calculation of measurement errors is described on P6L30. Errors were random and well-constrained, with little variability from sample to sample. We believe that adding error bars to Figure 6 (which contains 225 data points) does not add to the visual interpretation of these data. We did, however, add information about the measurement uncertainty to the figure caption.

P7L9: "How these method compare/differ from the derivation of the mass absorption efficiency presented in (Martins et al., 2009)?"

Martins et al., 2009 used a theory-based model to derive mass absorption efficiency, assuming an absence of multiple-scattering artefacts. Our goal was to link filter optical measurements to aerosol optical depth, and compare experimental findings with predictions from a radiative model of particles embedded in a multiple-scattering medium.
* * *

---

## Author Response (AR2)

1. Retrieve the 75 sample filters collected in the original experiment, and measure the reflectance and transmittance of each one on both sides: Rc and Rs, and Tc and Ts. If this effort would be difficult or uninformative for some reason, note such constraints in the paper to provide context for the choice of experimental design

2. Then replot Fig. S5 using these measured values rather the two-stream model outputs for
   a. ODs = ln((1-Rs)/Ts),
   b. ODc = ln((1-Rc)/Ts), and
   c. ODimprove = ln((1-Rc)/Tc).

   *The samples used in this study were collected over 3 rounds of experiments. Samples collected before 2017 (n=21) were no longer available for reanalysis. For the remaining 54 samples, Rc, Rs, Tc and Ts were measured. Due to the differences between Ts and Tc, we are no longer defining* **ODc = ln((1-Rc)/Ts)***, instead we use the appropriate definition* **ODc = ln((1-Rc)/Tc)***. For this subset of our samples, Fig. S5 was replotted and is now included as Fig. 7 in the mansucript. We found that ODs shows a stronger correlation with in-situ aerosol optical depth.*

3. Discuss how scattering by the filter substrate can yield Ts > Tc for measured values, even though equality is implied by the two-stream approximation.
   *This is mentioned in the manuscript on page 4, line 19 and page 7, line7.*

4. Explain what value was used for Tb when modeling ATN = ln(Tb/Ts) in Fig. S5. Was it the mean or median measured over a set of 20 blanks? Pre-sampling measurements on each of the 75 actual sample filters?
   *The mean Tb value for 20 lab blanks was used. This is now clarified in the Supplement and the manuscript (page 5, lines 16-19, 22-24).*

5. Discuss the variation of Tb observed in your individual blank filters, and compare it with that reported by White et al. (2016, Fig. 1) and Presler-Jur et al. (2017, Fig. 4).

   *The manuscript now includes the observed standard deviation in Tb. The variability is compared with that in the above references on page 3 of the Supplement. Overall, we find variability close to that observed for lab blanks in Presler-Jur et al., 2017 but lower than that for the field blanks in White et al., 2016.*

6. Calculate and plot the standard deviation to mean ratio of modeled filter OD measures over the observed range of Tb and Rb values, versus AOD. This is the analog to Fig. 3, looking at sensitivity to blank optics in place of sensitivity to SSA.

   *We felt that it would be difficult to interpret the separate information from those two analyses (sensitivity to SSA and blank optics) in a cohesive manner. Instead, we decided to redo Fig. 3 under a 'realistic blank' assumption. For each simulated sample (defined by a given SSA, τ$_{a,s}$ combination), we generated a Tb value from a normal distribution based on the observations from our 20 lab blanks. Then the three optical depth measures were calculated for the 5500 model samples, each associated with a distinct simulated blank. Standard deviations and means were then calculated as before. This resulted in some changes to the estimated values of the ratios of standard deviations to the means but the trend (variability: ODs<ODc<ATN) is largely unaltered.*

[revised manuscript text omitted]